# Prevalence and Trends of Physical Activity in Thai Children and Young People: Pooled Panel Data Analysis from Thailand’s Surveillance on Physical Activity 2012–2020

**DOI:** 10.3390/ijerph181910088

**Published:** 2021-09-25

**Authors:** Dyah Anantalia Widyastari, Pairoj Saonuam, Niramon Rasri, Kornkanok Pongpradit, Piyawat Katewongsa

**Affiliations:** 1Institute for Population and Social Research, Mahidol University, Salaya, Phutthamonthon, Nakhon Pathom 73170, Thailand; dyah.ana@mahidol.edu; 2Thai Health Promotion Foundation, Thung Maha Mek, Sathon, Bangkok 10120, Thailand; pairoj@thaihealth.or.th (P.S.); niramon@thaihealth.or.th (N.R.); 3Thailand Physical Activity Knowledge Development Centre (TPAK), Salaya, Phutthamonthon, Nakhon Pathom 73170, Thailand; kornkanok.p09@gmail.com

**Keywords:** physical activity, children, young people, pooled data, panel data, surveillance

## Abstract

This study aims to describe the level and trends of physical activity (PA) in Thai children and young people and examine PA changes during transitional periods. Employing nine rounds of Thailand’s Surveillance on Physical Activity (SPA) 2012–2020, this study pooled three sets of data and included children and young people aged 6–17 years in the analysis: 1595 in SPA2012–2016, 1287 in SPA2017–2019, and 853 persons in SPA2020. Face-to-face interviews were conducted in five regions, 13 provinces, and 36 villages in SPA2012–2019, whereas an online survey was administered in all provinces in SPA2020. The prevalence of sufficient moderate-to-vigorous PA (MVPA) among Thais aged 6–17 years ranged from 19.0 percent to 27.6 percent, with a significant drop during the period of COVID-19 spread in 2020. The average daily MVPA ranged from 46 to 57 min and dropped to 36 min during the pandemic. Boys were consistently more active than girls in all nine rounds of the SPA, and girls had more difficulty in maintaining or improving their PA level. A significant increase in the proportion of Thai children and young people with sufficient MVPA was observed during their transition from late primary to early secondary school grades.

## 1. Introduction

The benefits of regular physical activity (PA) for children’s health range from short-term fitness to long-term potential for reducing the incidence of chronic diseases that manifest in adulthood. Studies have documented that a regimen of 60 min of moderate-to-vigorous physical activity (MVPA) each day is positively associated with physical health status, fundamental motor skills [1,2], bone health [3,4,5,6], cognitive function [7,8,9], and socio-emotional development of children [9,10,11,12]. Nevertheless, despite the global effort to improve the health of young people, 80 percent of children and young people worldwide remain physically inactive or do not meet the recommended level of PA daily [13,14].

PA participation is a complex and multidimensional behavior that is determined by sociocultural, economic, and policy-related factors that operate across intra-/inter-personal, organizational, and environmental dimensions [15,16,17]. Although there is a great diversity of its influence on PA, generally, children from lower socioeconomic status (SES) households [18,19,20,21], girls [18,22], those with low parental/peer support [15,23,24,25], and those residing in an unsupportive/unsafe neighborhood [19,26] are less likely to obtain sufficient MVPA. Similarly, in the Thai context, the factors of age, sex, geographical location of residence/school, and parental/peer support also determined the PA level of Thai children and young people [27,28,29].

While there has been adequate research on the correlates of PA among children and young people, there is a lack of evidence—particularly in the Thai setting—concerning whether PA level is associated with periods of transition in a child’s life, particularly during school transition. Scholars from developing countries have reported that PA changes during school transition were common, and a greater decline was documented during the transition from primary to secondary school [30,31,32,33,34]. School characteristics, home environment (parents’ supports), and social space for PA was frequently reported as the correlates of PA changes during transitional periods [30,32].

The Thai government recognizes the importance of PA for the population at all stages of life and has accordingly launched various health promotion campaigns, starting in the early 2000s. There are policies, programs, and organized interventions to improve the PA level of young Thais. In addition, schools provide enabling and supporting inputs to encourage students to be more active. Nevertheless, only one in four Thai children and young people had sufficient MVPA during the period 2013–2017 [35,36,37]. The COVID-19 pandemic has undoubtedly disrupted and probably worsened children’s daily physical activities [38]. The closure of schools and public amenities have significantly reduced children’s opportunities for both structured and unstructured active play. The shift from onsite to online classes has required school young people to accumulate more screen time compared to pre-COVID-19 period [38,39,40].

This study aims to describe the level and trends of PA in Thai children and young people and examine PA changes over several transitional periods, including during the COVID-19 pandemic. In many settings, a low level of PA among children occurs as a result of adults’ poor understanding of biological and environmental changes. By understanding the timing and period of the decline or increase in PA, the school, family, and home community can create more opportunities to incorporate PA into a child’s daily life. The Thai government and policy makers can also use the results of this study as evidence in refining their strategies and designing appropriate interventions to improve PA of Thai children and young people and prevent or delay the onset of non-communicable diseases (NCD) of the population in the long run.

## 2. Materials and Methods

### 2.1. Data, Population and Sample

This study drew upon data from nine rounds of Thailand’s Surveillance on Physical Activity (SPA) 2012–2020 to enable analysis of a pooled set of data from two panels of individuals (SPA2012–2016 and SPA2017–2019) and analyzed data from SPA2020 as a special survey of the situation during the COVID-19 pandemic. The regular rounds of the SPA were designed as a longitudinal study and were jointly conducted by the Thailand Physical Activity Knowledge Development Centre, Institute for Population and Social Research of Mahidol University, and ThaiHealth.

In the first panel (SPA2012–2016), 1595 persons aged 6–17 years were enumerated. The sample was drawn from the general Thai population using multi-stage random sampling by considering the variance by geographical area (region and urban/rural), sex, and age. Face-to-face interviews were conducted in five regions, 13 provinces, and 36 villages, and the sample was nationally representative. A similar design and methods were applied in the second panel (SPA2017–2019) to select a nationally representative sample and follow-up of 1287 Thais aged 6–17 years. The two panels were independently sampled but derived from an identical sampling frame and technique to ensure national representativeness and enable data pooling. An additional sample in the subsequent year was caused by: (1) an increase in the structure of the population in the subsequent year, and (2) individuals that absented from the previous round returned as a sample.

The Thai government response to contain the spread of COVID-19 restricted people’s movement, and considering the fact that 85 percent of the Thai population had access to the internet, SPA2020 was conducted as an online survey. The online population sampling frame was derived from internet user data of the Thai National Statistical Office (NSO), classified by province. Probability random sampling was applied to draw a sample user of Facebook, with inclusion criteria of being a Thai citizen, aged 18–64 years, with a clear sex specification on their profile page, and who has internet access. Facebook was selected as the sampling platform since it provides the users’ location to enable multi-stage random sampling. In the first stage, we classified Facebook users by their area of residence (province) and then randomly selected two districts in each province. In the second stage, in each district, we sampled Facebook open groups and invited participants by systematic random sampling. The LimeSurvey web-based application was used for data collection.

The children and young people sample was drawn from parental response on living children and willingness for them to be involved in the study. Eligible respondents were then requested to respond to the young people PA questions together with their child(ren). A total of 853 children and young people aged 6–17 years were included in the analysis. Of the total sample in the analysis, the proportion of boys was slightly higher (51.2 percent) than that of girls (48.8 percent). Classified by educational level, the proportion of participants who attended early secondary school was the highest (31.9 percent), followed by late secondary (29.1 percent) and late primary school. More of the participants resided in an urban area, and predominantly lived in the central or northern regions (Table 1).

### 2.2. Measurements

We used a 24 h activity log (clock-like record) to assess PA (Figure 1). We asked the study participants whether, in a typical week, their or their children’s activities involved any bodily movement that requires energy expenditure in moderate or vigorous intensity of PA. We recorded the activities on an hourly basis and divided these into minutes when there was an activity change. The duration of activities in all domains and intensities (expressed in minutes) was then multiplied by the number of active days and summed as total active time weekly. As the final measure, PA was expressed in average moderate-to-vigorous PA (MVPA) daily. Sufficient PA was defined based on the WHO recommendation on PA for children and young people: (1) sufficient, if a child achieved at least 60 min MVPA daily; and (2) insufficient, if a child achieved less than 60 min MVPA daily [41].

PA transition was defined as changes in PA level in terms of the following: (1) changes in the proportion of children and young people who met the sufficiency level; and (2) changes in average daily MVPA. We observed the same individuals for three years (2017–2019), documented the average daily MVPA, and expressed the result in minutes. The children/young people who had sufficient MVPA were categorized into the following: (i) maintained sufficient; (ii) improved (insufficient to sufficient); (iii) declined (sufficient to insufficient); or (iv) remained insufficient.

Sociodemographic characteristics of respondents were assessed to portray the patterns of PA by different groups of young people. Sex was differentiated into (1) male and (2) female, whereas age was recorded as a single age, ranging from 6 to 17 years. We also categorized PA by the participant’s school grade level, whether they were currently enrolled in (1) early primary school (grades 1–3); (2) late primary school (grades 4–6); (3) early secondary school (grades 7–9); or (4) late secondary school (grades 10–12). Geographical location was defined based on region of residence and coded as follows: (1) central; (2) north; (3) northeast; (4) south; or (5) Bangkok. Participants were also classified by residence in an (1) urban (municipality); or (2) rural (non-municipal) area.

### 2.3. Data Analysis

Although different data collection methods were applied (i.e., face-to-face versus online), several adjustments were made to ensure comparability of the data. A test–retest procedure was performed by exposing 30 individuals to paper and online questionnaires in order to compare response, validate the data collection instruments, and ensure measurability. The comparability of the two methods was confirmed by the Bland–Altman [42] coefficient (mean difference = 0.16, *t* = −0.026) and Pearson’s correlation coefficient (0.882), as reported in our previous paper [43]. A paired *t*-test was used to evaluate each individual item of the questionnaires, and the analysis found no significant difference between offline and online response. Sampling bias was controlled by testing samples that were included and excluded in the analysis.

We pooled all participants aged 6–17 years from SPA2012–2020. Two PA trajectories were observed: (1) by the survey year, to examine trends in the prevalence of sufficient MVPA among children and young people during 2012–2020; and (2) PA transition, by observing the changes in the proportion of young people who had sufficient MVPA in the most recent panel (2017–2019) before the pandemic period. We used IBM SPSS Statistic version 25 (IBM Corp, Armonk, NY, USA), licensed to Mahidol University, for data analysis. Pearson chi-square was used to test whether there was a statistically significant difference in the PA transition by sex, residential area, and region. ANOVA and *t*-test were employed to examine the difference in the average daily MVPA based on participants’ characteristics (i.e., sex, educational level, residential area, and region). Multiple regression analysis was performed to determine the correlates of PA transition among SPA2017–2019 samples.

## 3. Results

### 3.1. Physical Activity of Thai Children and Young People: Level and Trends

The prevalence of sufficient MVPA among this sample of Thais aged 6–17 years ranged between 19.0 percent (SPA2020) and 27.6 percent (SPA2014). The lowest prevalence of MVPA occurred during the COVID-19 pandemic period, when stringent containment and control measures were imposed nationwide, starting in March 2020. However, even during the pre-pandemic period (2012–2019), the prevalence of Thai children and young people engaging in 60 min of MVPA daily was generally low (Table 2). The average daily MVPA also showed a fluctuating trend, from 46 min in SPA2015 as the lowest point, to 57 min in SPA2016. There was a significant drop in PA during 2020, when children and young people collected only 37 min of MVPA on average (Figure 2).

An irregular pattern was also observed in the proportion with sufficient MVPA by area of residence. The proportion of children and young people with sufficient MVPA who lived in a rural area was slightly higher compared to their urban counterparts. Based on region of residence, most children/young people who resided in Bangkok were less active than those who lived in the Central or North regions. Similarly, the proportion of young people living in the Northeast who engaged in sufficient MVPA was higher than those who resided in the South or Bangkok (Table 2).

The proportion of Thais enrolled in an early primary school grade (aged 6–8 years) who had sufficient MVPA was lower than those in the late primary grades (aged 9–11 years) in the SPA2012–2016. However, this trend was reversed in SPA2017–2020. Comparing the level of MVPA sufficiency between the four school grade levels, this study found that secondary school students were generally more active than their younger counterparts. The proportion with sufficient MVPA was the highest among early secondary school students in SPA2014–2017, while late secondary school students were more active in SPA2018–2020 (Table 2).

### 3.2. Physical Activity Transition among Thai Children and Young People

A transition in PA of Thai children and young people in this study was observed from the changes in the proportion who met the recommended sufficient level and changes in the average daily MVPA. A change in the proportion of sufficient MVPA was observed during the transition period from late primary school (22.0 percent) to early secondary school (25.9 percent). While there has been a relatively stable proportion of primary school students with sufficient MVPA, the level of PA increased slightly by school grade level (Figure 3).

Observing the latest panel only (SPA2017–19), the vast majority of Thai children and young people had an insufficient level of MVPA during three years of observation (2017–2019). The proportion of young people who maintained their PA at/above the recommended level of MVPA of 60 min daily was low, and this pattern is consistent for all regions. Although lacking statistical significance, young people who resided in Bangkok or the central region led their peers in maintaining sufficient MVPA; the proportion of those who had improved or declining MVPA was similar with their counterparts in the other regions. Most of the children who lived in the south region had a declining transition (from sufficient to insufficient MVPA), whereas those who resided in the northeast remained mostly at an insufficient level of MVPA (Table 3).

Residential area (urban/rural) also was not significantly correlated with the changes in the proportion of PA sufficiency. Both urban and rural residents had a similar pattern of PA, with the highest proportion remaining at an insufficient level. However, the proportion of rural children who could maintain their PA at a sufficient or improved level was slightly higher than their urban counterparts (Table 3).

While there was no significant association between residential area to the changes in the proportion of PA sufficiency (Table 3), there was a significant difference in the average daily MVPA collected by urban and rural children in SPA2017 (Table 4).

Educational level was not significantly associated with PA transition (Table 3). At all levels, the proportion of children and young people who remained at an insufficient level was the highest, compared to those who could improve or maintain sufficient MVPA (Table 3). A significant difference (*p*-value = 0.019) in the average daily MVPA was also found among children by school grade in 2017 (Table 4).

PA transition—both in the proportion of sufficient and average daily MVPA—was significantly correlated with sex. The proportion of boys who remained at an insufficient level of MVPA (46.2 percent) was the highest compared to those who improved or maintained PA sufficiency. The proportion of girls with no improvement in PA (remained at an insufficient level) was much higher, at 64.2 percent. The proportion of boys and girls who could maintain their PA level at/above the WHO-recommended 60 min daily (12.9 and 4.4 percent, respectively) was the lowest among transition groups (Table 3). The average daily MVPA of boys was also consistently higher than of girls during the three years of observation (Table 4).

Since there was no significant association of the study covariates (except sex) with transition in sufficiency level, multiple regression analysis was performed to test the effect of sex, educational level, region, and residential area on average daily MVPA of SPA2017–2019′s samples. The analysis found that sex, educational level, and residential area were significantly associated with change in average daily MVPA of children and young people, and that each variable can independently predict change in average daily MVPA when the other variables are held constant. The conditional effect of sex is stronger than that of educational level and residential area, but in total, the predictors can only account for approximately 11 percent of the overall variation in the outcomes (Table 5).

## 4. Discussion

On average, only one in four Thai children and young people met the recommended level of 60 min of MVPA daily throughout 2012–2020. The prevalence of sufficient MVPA was at its lowest level during the period of spread of COVID-19 in 2020 and highest in 2014. The average daily MVPA also fluctuated between 46 and 57 min, with the lowest (37 min) in 2020.

It cannot be denied that COVID-19 has disrupted people’s lives, including reducing the opportunity to engage in PA with others. Although indoor PA (‘Fit from Home’) has been promoted, the vast majority of Thais experienced a significant drop in the frequency and level of PA [44]. The PA level of Thai children and young people was also at its lowest point during 2020 and was significantly below the baseline level in 2012. With the periodic closure of schools and public amenities, Thai children and young people have lost their major PA opportunities. The restrictive COVID-19 control measures have confined young people to their homes and immediate neighborhoods, and online teaching further requires school-age young people to be sedentary for many hours of the day. The COVID-19 pandemic has also contributed to a decline in PA of young people globally. Around the world, young people are experiencing a decline in all types of PA, except for that related to performing household chores [39,45,46].

The low level of Thai children and young people with sufficient MVPA prior to the pandemic suggests that past policy and practices have been inadequate or have not addressed the root causes of exercise behavior among the younger generation. Although the national policy has targeted reduction in childhood obesity to under 10 percent and recommends PA as one of the principal interventions to achieve that goal [47,48,49], there are no national guidelines for promoting PA among children and young people to show how parents, guardians, and teachers can intervene or for optimizing PA opportunities for children and young people. With a further decline in PA levels during the pandemic, there is no doubt that more interventions, programs, and policies should be implemented in order to improve the PA of young Thais.

Sex was significantly associated with PA level and transition. Boys were consistently more active than girls, and that finding is consistent with global patterns of PA by sex [41]. By following more than 600 young people from 2017 to 2019, this study also found that girls had greater difficulty maintaining or improving their PA level. The sex-related difference in the PA transition could be explained by physical development theory, which argues that a boy’s development increases his bone and muscle mass to facilitate PA, while puberty for a girl increases fat mass that reduces her ability to be physically active [50]. For Thai children and young people, the low participation of girls in PA is also culturally constructed, since most organized sports are geared more toward males. The Thai cultural norm is that the proper girl should be calm and neat (i.e., not making abrupt movements or being sweaty), and that discourages them from engaging in MVPA [36]. The belief in the importance of fair skin also has been suggested as a significant barrier for Thai girls to engage in outdoor PA, especially if they have to be exposed to direct sunlight [51]. Furthermore, Thai public schools require students to wear uniforms, and the standard girl’s school skirt is not conducive to vigorous movement, compared to the boy’s uniform, which is knee-length shorts [36].

Geographical region was not correlated with both types of PA transition during 2017–2019. This finding suggests that, throughout Thailand, most Thai children and young people failed to improve or maintain their PA level. Indeed, most fell into, or remained at, an insufficient level. The insignificance of region with regard to the PA transition is possibly associated with the Thai policy of decentralization, which has been implemented since the early 2000s. That policy authorized local government to manage its own revenue collection and spending in order to increase income equality, promote self-reliance, and encourage grassroots problem-solving [52]. The decentralization policy also includes developing areas for promotion of PA. Local playgrounds and public parks were now the responsibility of municipalities in urban areas and of sub-district administrative organizations in rural areas. The insignificance of region in predicting PA transition implies equality of resources between regions, particularly with the availability of national standards for a safe-built environment and infrastructure. Indeed, the gap in PA between regions seems to be narrowing as reflected by the same irregularity of patterns in the average daily MVPA and the prevalence of sufficient MVPA.

Residential areas were significantly associated with change in average daily MVPA during 2017–2019. While the development of infrastructure and the built environment in Thailand aims at reducing urban–rural differences generally, it cannot be denied that the opportunities for urban dwellers to engage in MVPA remain deficient. Unsafe environment was often pointed to as the major reason for low PA level among urban children and young people in Thailand [36].

This study found that Thai students at the primary school level were the least active. That finding is consistent with previous studies in Thailand which found that primary school students aged 6–11 years had generally lower probability of meeting the recommended level of 60 min of daily MVPA, compared to their older counterparts [35,36,37]. However, the global pattern is the reverse, whereby the rate of meeting the PA guideline decreases with age and school grade level [53,54,55,56,57].

The lower level of PA among primary school students could be attributed to policies and practices in primary schools that currently prioritize infrastructure development at the expense of facilities for PA. While most schools have policies to support PA equipment and sports facilities, only about one-third of schools have a recess policy [58]. What is more, the standard Thai primary school curriculum prescribes an average of only 40 min of physical education per week, in contrast with a long duration of daily classroom learning (from 8:00 a.m. to 3:30 p.m.), which is predominantly in the sitting position. The Thai government recognizes that the confinement of children to the classroom is a problem and has introduced a “Teach Less Learn More” policy. Unfortunately, this policy does not replace classroom sitting with activities that promote PA. Organized PA is implemented in most schools, but the PA lacks quality and guidelines to ensure uniformity [36]. Primary school students also do not have adequate opportunities for PA after school, in part because parents would prefer their child to have additional (private tutorial) classroom time outside the school. On the other hand, high school students have more opportunity for active travel and sports after school based on their interests [59,60]. Thai adolescents also use sports as a means of socialization with their peers, as reflected by the higher proportion who engage in sufficient MVPA if they have friends who do so [28].

Although the longitudinal study with panel-data design is a strength of this study, several limitations should be acknowledged. First, due to the COVID-19 pandemic, the 2020 sample was not drawn from the same sampling frame of the previous survey rounds. Second, PA assessment using self-reported activity log is a subjective measure and may have led to under- or over-estimation in children’s PA level due to children’s and parents’ recall error. It is true that there were considerable variabilities between subjective and objective measures of PA, particularly among the younger population [61]. However, this study was conducted as a national surveillance system that involved multiple years of large-scale samples, and using an accelerometer to measure PA would have been impractical. In addition, the instrument of this study has undergone a reliability test, where the responses between subjective (activity log) and objective (accelerometer) measures were compared. The Pearson’s correlation and Bland–Altman coefficient resulted in an acceptable level, suggesting that the instrument was reliable to measure the PA levels of Thai children and young people. Third, this study is lacking several important variables such as school characteristics and parental socioeconomic status (SES). Fourth, any change in PA seen between 2016 and 2017 and 2019 and 2020 may reflect the change in sample rather than actual population-based changes. However, with panel data collected over time and among the same individuals, trends or transitions of behavior can be observed, while variability between subjects can be controlled. The same instruments and protocols were also employed to assess PA constructs in different point of times to ensure that the changes in PA level over time are attributable to actual changes in the constructs.

## 5. Conclusions

On average, only one in four Thai children and young people met the WHO-recommended level of 60 min daily MVPA over the period of study. The prevalence of sufficient MVPA fluctuated, showing a significant drop during the spread of COVID-19 in 2020. Primary school students and girls consistently had a lower level of PA compared to boys and secondary school students. PA changes were obvious during the transition from primary to secondary school, and sex was significantly correlated with these changes. More than boys, girls experienced greater difficulty in maintaining or improving PA level during 2017–2019.

Considering the low level of PA among Thai children and young people, particularly with the significant drop due to COVID-19, programs and interventions should be directed toward a gradual increase in MVPA, with a focus on the primary school grades. National guidelines on PA promotion are required to ensure a whole-system approach that involves schools, the home community, and family in order to help the entire society move in the same direction. Specific programs should be designed to help girls and younger children overcome any physical, environmental, and cultural barriers by providing more opportunities for PA at school and at home.

## Figures and Tables

**Figure 1 ijerph-18-10088-f001:**
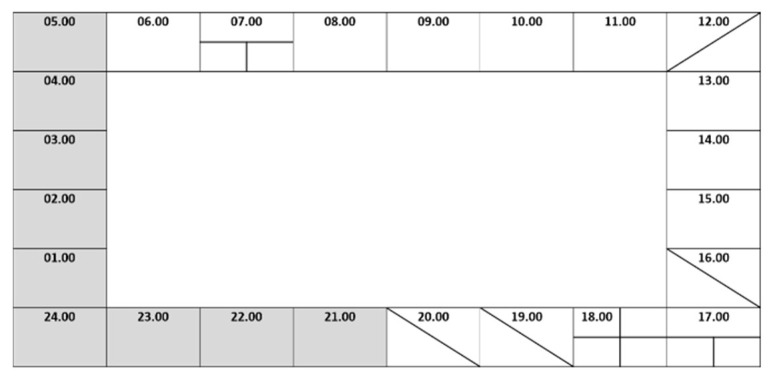
24 h activity log.

**Figure 2 ijerph-18-10088-f002:**
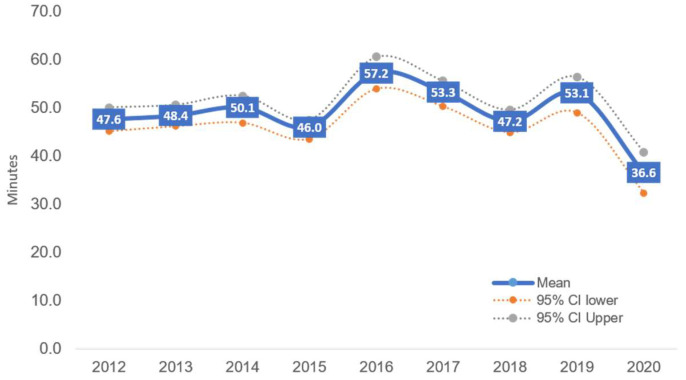
Average daily MPA (in minutes) among Thai children aged 6–17 years. Boys were consistently more active than girls in all nine rounds of the SPA. Although the pattern is inconclusive, the proportion of boys who met the recommended level of 60 min of daily MVPA increased in the most recent rounds (SPA2018–2019), but there was a sharp decline in SPA2020 during the spread of COVID-19. The proportion of girls with adequate MVPA ranged between 13.4 percent (SPA2020) and 23.4 percent (SPA2012) (Table 2).

**Figure 3 ijerph-18-10088-f003:**
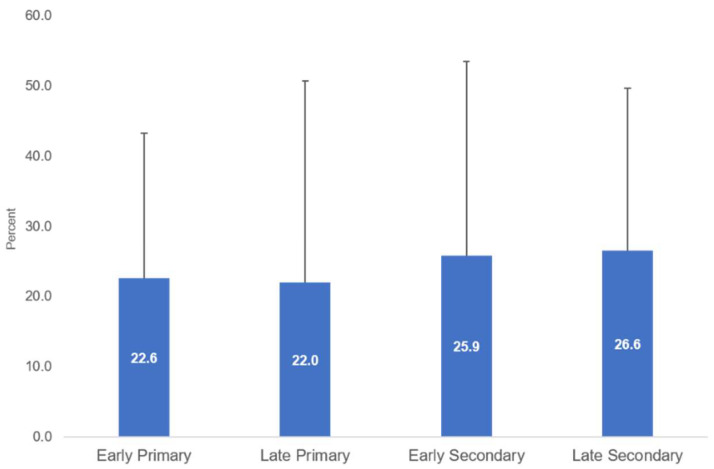
Average percentage of sufficient MVPA of Thai children in 2017–2019, classified by education.

**Table 1 ijerph-18-10088-t001:** Characteristics of the sample.

	*n*	(%)
Sex		
Male	6332	51.2
Female	6035	48.8
Educational Level (age in years)		
Early Primary (6–8)	1794	14.5
Late Primary (9–11)	3021	24.4
Early Secondary (12–14)	3949	31.9
Late Secondary (15–17)	3603	29.1
Residential Area		
Urban	6680	54.0
Rural	5687	46.0
Region		
Bangkok	1303	10.5
Central	3370	27.2
North	3278	26.5
Northeast	2188	17.7
South	2228	18.0

**Table 2 ijerph-18-10088-t002:** Prevalence of sufficient MVPA among Thai children and young people: 2012–2020.

Children’s Characteristics	Percentage Sufficient Physical Activity (at Least 60 min Daily) (*n* = 12,367)								
	2012	2013	2014	2015	2016	2017	2018	2019	2020
	(*n* = 1595)	(*n* = 1637)	(*n* = 1493)	(*n* = 1480)	(*n* = 1404)	(*n* = 1287)	(*n* = 1320)	(*n* = 1298)	(*n* = 853)
Sex									
Male	27.8	24.2	29.1	25.8	29.4	28.6	31.4	31.0	20.3
Female	23.4	17.4	22.4	19.4	21.4	20.5	22.9	21.6	13.9
Educational Level (age in years)									
Early Primary (6–8)	24.9	18.4	25.5	21.1	21.4	25.5	24.4	24.6	17.7
Late Primary (9–11)	22.4	19.1	26.4	23.2	23.9	24.2	21.9	21.7	15.2
Early Secondary (12–14)	24.7	23.9	28.2	28.2	28.3	27.2	27.4	23.0	22.0
Late Secondary (15–17)	27.8	27.9	27.2	26.2	27.1	24.0	30.9	28.4	19.7
Residential Area									
Urban	23.5	20.8	26.8	22.1	23.3	25.6	25.4	24.3	18.6
Rural	26.7	21.3	27.8	25.4	30.1	24.9	28.8	24.5	20.0
Region									
Bangkok	22.8	18.0	21.9	27.4	20.9	27.3	24.2	25.0	18.5
Central	29.2	23.1	31.3	27.7	28.1	25.9	23.6	27.2	14.7
North	26.7	24.7	25.5	25.2	24.7	21.6	29.6	20.4	19.9
Northeast	24.6	21.0	29.8	22.4	28.3	28.1	29.7	22.0	23.9
South	29.1	20.6	22.2	19.7	27.9	24.0	26.3	26.6	17.5
Total (Thailand)	24.9	20.9	27.6	23.2	26.4	25.3	26.2	24.4	19.1

**Table 3 ijerph-18-10088-t003:** Physical activity transition among Thai children and young people 2017–2019.

Children’s Characteristics	Percent PA Transition in 2017 and 2019 (*n* = 628)					
	Maintained Sufficiency	Improved	Declined	Remained Insufficient	Pearson Chi-Square	*p*-Value
	(Sufficient to Sufficient)	(Insufficient to Sufficient)	(Sufficient to Insufficient)	(Insufficient to Insufficient)		
Sex						
Male	12.9	20.7	20.1	46.2	53.412	0.000
Female	4.4	18.8	12.5	64.2		
Educational level (age in years)						
Early primary (6–8)	16.4	10.9	20.0	52.7	10.241	0.331
Late primary (9–11)	7.0	16.7	17.7	58.6		
Early secondary (12–14)	6.5	16.0	17.5	60.0		
Late secondary (15–17)	11.2	13.9	13.4	61.5		
Residential area						
Urban	8.8	14.0	15.4	61.8	2.178	0.536
Rural	9.0	16.6	18.1	56.3		
Region						
Bangkok	10.3	17.2	15.5	56.9	17.458	0.133
Central	11.8	15.7	12.4	60.1		
North	4.3	20.9	18.7	56.1		
Northeast	8.4	15.7	12.0	63.9		
South	9.7	19.7	20.5	50.0		

**Table 4 ijerph-18-10088-t004:** Average daily MVPA among Thai children and young people 2017–2019.

	SPA2017						SPA2018						SPA2019					
	Mean	SD	Lower	Upper	F/*t*-Test	*p*-Value	Mean	SD	Lower	Upper	F/*t*-Test	*p*-Value	Mean	SD	Lower	Upper	F/*t*-Test	*p*-Value
	(Mins)		(Mins)	(Mins)			(Mins)		(Mins)	(Mins)			(Mins)		(Mins)	(Mins)		
Sex																		
Male	51.7	52.8	46.0	57.4	6.424	0.000	52.4	42.7	47.8	57.0	6.701	0.000	39.7	38.2	35.6	43.8	7.494	0.000
Female	30.2	29.1	26.9	33.5			32.2	32.7	28.4	35.9			21.0	23.1	18.4	23.7		
Educational level (age in years)																		
Early primary (6–8)	46.8	40.3	35.9	57.7	3.323	0.019	48.4	37.0	38.4	58.4	0.542	0.654	34.1	31.3	25.6	42.5	0.194	0.90
Late primary (9–11)	34.8	38.9	29.2	40.4			41.0	38.4	35.5	46.6			31.1	28.8	26.9	35.2		
Early secondary (12–14)	40.0	34.9	35.2	44.9			42.3	36.9	37.2	47.4			30.4	33.3	25.8	35.1		
Late secondary (15–17)	48.6	57.7	40.2	56.9			43.9	44.3	37.5	50.3			30.4	38.0	24.9	35.8		
Residential area																		
Urban	38.0	39.0	33.9	42.1	−2.22	0.027	40.8	38.8	36.8	44.9	−1.49	0.136	30.3	33.8	26.7	33.8	−0.55	0.584
Rural	46.2	50.5	40.2	52.2			45.6	40.6	40.8	50.4			31.7	32.8	27.9	35.6		
Region																		
Bangkok	37.2	43.7	25.8	48.7	0.864	0.485	40.8	39.5	30.4	51.2	1.857	0.116	32.9	34.4	23.9	42.0	1.821	0.123
Central	40.8	41.5	34.2	47.5			42.1	36.7	36.2	48.0			34.4	39.5	28.1	40.7		
North	42.2	55.4	32.9	51.5			44.7	47.2	36.8	52.6			30.6	32.1	25.2	35.9		
Northeast	35.9	37.0	27.9	44.0			52.5	45.9	42.5	62.5			35.0	35.2	27.3	42.7		
South	45.5	41.5	39.6	51.4			38.9	32.0	34.3	43.4			26.1	26.9	22.3	29.9		

**Table 5 ijerph-18-10088-t005:** Determinants of change in average daily MVPA among Thai children and young people 2017–19.

Variable	B	*p*-Value	95% C.I. for EXP(B)	
			Lower	Upper
Sex				
Male (Ref.)				
Female	−20.149	0.000	−25.473	−15.365
Educational level (age in years)				
Late secondary (15–17) (Ref.)				
Early primary (6–8)	−2.131	0.667	−11.858	7.597
Late primary (9–11)	−8.220	0.014	−14.745	−1.696
Early secondary (12–14)	−6.508	0.046	−12.910	−0.105
Residential area				
Urban (Ref.)				
Rural	5.489	0.043	0.166	10.812
Region				
Bangkok (Ref.)				
Central	−1.111	0.826	−11.022	8.800
North	−1.825	0.715	−11.651	8.001
Northeast	−6.076	0.287	−17.285	5.132
South	−3.161	0.541	−13.320	6.998
Constant	67.183	0.000	53.747	80.619
R^2^	0.112			
F	8.619	0.000		
df	9			
Number of observations	627			

## Data Availability

SPA data are available in TPAK repository, https://tpak.or.th/?p=4151 (accessed on 25 January 2021).

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
