# Peer review of "Prevalence and Trends of Physical Activity in Thai Children and Young People: Pooled Panel Data Analysis from Thailand’s Surveillance on Physical Activity 2012–2020"

_ijerph, 2021, doi:10.3390/ijerph181910088_

Round 1
Reviewer 1 Report
Please see the attached document.

Author Response
Thank you for your constructive comments to improve our manuscript.
Please find our responses in the file attached.
Best regards,
Piyawat

Reviewer 2 Report
Thanks for this interesting manuscript. In its’ current form, I have some concerns regarding whether it is suitable for publication. The introduction does not tell the story in such a way that the reader is left understanding why the study needs to be conducted. The introduction talks a lot about health outcomes and correlates, and yet no health outcomes or modifiable correlates are assessed in the study. I also see that there are issues with the way in which the data was pooled for analysis. There is very little detail regarding the methodological considerations and how/why the data analysis was conducted, which leaves the reader guessing as to what was done and why. Please see below for more specific feedback.
Introduction:
Line 29 – what type of physical activity? Ensure that it is clear that this statement is referring to moderate-to vigorous-intensity aerobic physical activity.
Line 34 – okay, so here MVPA is mentioned. Switch it around to note this on line 29, then on line 34 MVPA can be used.
Lines 37-38 – children from high SES households are more active in many high-income countries, so making this statement and referring only to a small study conducted only in Portugal is not entirely accurate or representative of all high-income countries. Note also that Thailand is considered to be an upper-middle income countries, so perhaps it is best that this evidence is reflective of other comparable countries.
Lines 43-45 – there are a number of studies, including reviews, regarding how physical activity changes during key transition periods (e.g. childhood to adolescence, primary to secondary school etc.) which I would encourage the author to read and refer to here.
Lines 55-57 – higher sedentary time or higher screen time? I think screen time is the right terminology to use here. Children typically spend as much as 75% of their school day sitting in the school setting, that has changed very little during the pandemic, but a much greater proportion of the school day is spent using screens while sitting.
Lines 57-68 – the shift back to health outcomes seems out of place here. I suggest a full restructuring of this introduction to ensure that it flows logically and leads the reader towards the aims/objectives.
Materials and methods:
I am not a statistical expert however I see major issues with pooling the data collected across 9 years considering the same participants completed all surveys from 2012-2016 and 2017-2019 (i.e. longitudinal cohort). Any change in PA seen between 2016-2017 and again from 2019-2020 may be a reflection of the change in sample rather than actual population-based changes.
How were there more participants in 2018 and 2019 compared to 2017 if the sample were recruited in 2017?
Lines 162-163 – what was the PA transition that was analysed here? As it stands, there is very little explanation of this. As I read further down, I can now see that this is clarified under second 3.2 (results). Please ensure this information is included in the methods section and relevant evidence from existing literature is introduced in the introduction.
Results:
Page 7, lines 2-7 – was this proportion change between late primary and early secondary statistically significant? What statistical test was conducted to determine the magnitude of change?
Consider re-ordering the reporting of results from tables 3 and 4. I suggest report results regarding characteristic differences by transitions (table 3) first, and then characteristics differences by average daily MVPA within each time point (table 4) separately. Also, why is there no breakdown by timepoint for the data collected from 2012-2016 like there is for 2017-2019?
Discussion:
Lines 44-45 – Declines in PA are happening for youth ‘around the world’ and yet only one study from Canada is cited. Please include evidence from additional countries to support the statement being made.
Lines 46-48 – do you mean the low levels with sufficient MVPA across ALL YEARS or just during COVID-19? Without a timeframe, it isn’t clear, and this paragraph follows the paragraph related to COVID times.
Lines 48-49 – this statement refers to the national policy, and as such this policy should be cited.
Lines 61-63 – this is a really pertinent point and rings true of many cultures. It should be supported by evidence.
Lines 95-99 – Not sure about this explanation. PA levels at an earlier age has been shown to be associated with PA levels at later age (Thailand as an exception of course). i.e. higher PA during childhood = higher PA during adolescence.
Lines 112-115 – Could it also be the case that adolescents are more freely able to actively travel independently, whereas younger children do so less? Not being from Thailand myself, I’m not sure but this is something that is commonly seen in other Western cultures.
Line 116-117 – what is it about the research design and methodology that can be seen as a strength?
Author Response
Thank you for the comments and suggestions.
Please find our responses attached.
Best regards
Piyawat

Round 2
Reviewer 1 Report
Measuring MVPA via self-report versus accelerometry produces considerably different results in a sample of young adolescents. Future studies should use caution when selecting outcome measures if the intent is to identify modifiable correlates of MVPA in youth. The authors just recorded MVPA with a 24-hour activity log (clock-like record) to assess PA. This method is totally subjective and many studies have highlighted the importance of providing comparative objective measures. Please see for example the paper entitled: Correlates of subjectively and objectively measured physical activity in young adolescents.
You should really discuss this in the discussion section. How this method used can affect your results? Also state this in the limitation section.
As I said you did not include parental SES which is an essential information. You should state this in the discussion and limitation section.
The discussion section must be improved.
Author Response
Thank you for your suggestion.
Please kindly find our responses attached.
Best regards
Piyawat

Reviewer 2 Report
Thank you for providing your revised manuscript and responses to my previous comments.
Below are some points that are still yet to be adequately addressed:
- As per my previous suggestion, the authors have not included reference to or discussion about how physical activity has been shown to change during key transition periods (e.g. from primary to secondary school). This should be included in the introduction as that is a key focus point of the analysis/results for this manuscript. The current manuscript does not look at modifiable correlates or determinants of changes in physical activity, so the content related to that in the introduction currently could be replaced with more pertinent content.
- My comment regarding the fact that any change in PA seen between 2016-2017 and again from 2019-2020 may reflect the change in
sample rather than actual population-based changes still remains. I suggest that this be discussed as a limitation of the study. - Thank you for clarifying why the numbers differed in the 2017-2019 samples. A mention of this should be included in the methods section of the manuscript.
Author Response
Thank you for constructive comments
We have revised following your suggestions.
Best
Piyawat
